# Improvement of Fire Safety Performance for Nursing Homes by Using Fireproof Curtains with a Water Film System

Su-Hua Chiu [1], Charlene Wu [2], Chen-Yu Chen [1,*], Ta-Hui Lin [2,3] and Shin-Ku Lee [2,*]

1 Department of Architecture, National Cheng Kung University, Tainan 701401, Taiwan
2 Research Center for Energy Technology and Strategy, National Cheng Kung University, Tainan 701401, Taiwan
3 Department of Mechanical Engineering, National Cheng Kung University, Tainan 701401, Taiwan
* Correspondence: chenyu88@mail.ncku.edu.tw (C.-Y.C.); sklee1015@gmail.com (S.-K.L.)

**Abstract:** Nursing home residents often have impaired faculties, which limit their evacuation responses in light of accidents. Specifically, fire accidents can pose significant risks to both residents and property. In this study, we designed a water film cooling system to enhance the heat and smoke resistance of a fire retardant curtain without a fire resistance rating. The experimental results on a full-scale door/wall refractory furnace and smoke-barrier system showed that the proposed curtain had 1.5-h heat resistance properties, while the smoke control performance could meet the ISO 5925-1 requirement. Finally, we used Fire Dynamics Simulator (FDS) 6.0 to simulate an existing nursing home with this type of fireproof curtain setup for horizontal evacuation waiting space. The results show that our proposed water film cooling system can effectively inhibit the diffusion of smoke during the initial stage of a fire, creating a safe evacuation waiting zone for disadvantaged evacuees.

**Keywords:** nursing home; fire retardant curtain; water film system; heat resistance properties; horizontal evacuation waiting space

## 1. Introduction

### 1.1. Risks Posed to Long-Term Care Facilities in Taiwan

With improvements in both the social welfare and the quality of health care in Taiwan, life expectancy has increased steadily; the elderly population now accounts for 11.5% (2.69 million) of the total population. Furthermore, it is estimated that those over 65 years old will exceed 20% of the total population by 2025 [1]. This trend indicates that Taiwan is one of the fastest aging countries in the world. As the aging population in Taiwan continues to rise, so too does the number of nursing homes, which reached 1021 by 2007—a 17.6-fold increase compared to the number of facilities registered in 1995 [2].

Although residents in long-term care facilities often include the elderly, these establishments also house those with limited mobility or those who need long-term nursing care. Additionally, people with physical and/or mental disabilities may have slowed actions and reflexes and can thus exhibit impaired evacuation behavior during a fire. For example, the average walking speeds of a freely moving elderly person, a male using crutches, and a female using crutches are in the range of 0.63 and 1.35 m/s; all of which are slower compared to that of a healthy younger person (1.4 m/s) [3]. Therefore, it is evident that fire incidents pose significant risks to nursing home residents.

Table 1 details past major fire accidents in nursing homes. The fire at the nursing home at Hsinying Hospital in 2012 resulted in the loss of 12 lives and 60 more injured. Post-incident analysis concluded that the fire was not able to be contained and extinguished due to the lack of an automatic sprinkler in the storage room [4]. Other reasons include too few staff during the nightshift and the lack of fire detector central control monitoring. Finally, the patients were not mobile enough to escape, and thus heavy casualties resulted.

**Table 1.** The cases of fire occurring in nursing homes in Taiwan.

| Time of Occurrence | Facility Location | Fire Location | Cause of the Fire | Deceased (Injured) | Reasons for Major Damage |
| --- | --- | --- | --- | --- | --- |
| 15 January 1988 | New Taipei City | NA | Appliances overheating | 11 (10) | The fire-fighting facilities of this center were seriously substandard. There was only one escape exit, and the other passage was sealed. In the center, there was only one fire extinguisher with no hydrant and the layout of the beds was dense. The building space did not have a smoke control design, resulting in a lot of smoke that caused choking. |
| 19 October 2004 | Keelung City | Heater | Appliances overheating | – | A heater in the Ren-de building in the Ren-ai Nursing Home of Keelung City caught fire. |
| 15 September 2005 | Kaohsiung City | NA | Lighter | – | A resident at the Sung Chiao Home for the Aged had possession of a lighter and caused a fire. |
| 23 October 2012 | Tainan City | Storage room | Arson | 12 (60) | A patient lit tissue paper and threw it into an unlocked storage room full of clothes, which caused the fire. |
| 27 May 2014 | Tainan City | Storage room | Arson | 0 (12) | A Vietnamese female worker committed arson due to pressure at work and homesickness. |
| 6 July 2016 | New Taipei City | Storage room | Short circuit | 5 (29) | The short circuit of electrical application resulted in igniting the stacked goods in the storage room |
| 10 March 2017 | Taoyuan City | Ward | Candle fire | 4 (13) | The power system was under maintenance in the early morning. Thus, the caregiver used candle lighting; but the candle overturned and resulted in an accidental fire |
| 19 May 2017 | Pingtung County | Ward | – | 4 (56) | Smoke spread from the broken glass and wooden door of the ignited ward and into the corridor. Meanwhile, the fire detector systems, smoke exhaust system, and sprinkler were not activated during the fire. |
| 13 August 2018 | Taipei City | Ward | Short circuit | 15 (14) | The short circuit of an electrical appliance resulted in igniting the mattress and curtain; smoke spread from the open door of the ignited ward and into other wards. |

### 1.2. Fire Safety Improve Schemes of Nursing Home

As a response to improve the hazard prevention and evacuation capabilities of these establishments, the Taiwanese government drafted and passed the Enhance Public Safety of Long-term Care Facility Act in 2017. This statute outlines that building firefighting facilities should include an automated sprinkler system, encourage the use of flame resistant materials throughout the facility, and ensure the installation of fire safety equipment such

as fire hydrants. Other improvements include but are not limited to up-to-date standards regarding the fire facility and equipment inspection. The aforementioned improvements were drawn from many studies, as these investigations noted the use of flame-retardant or non-combustible materials, the addition of early smoke detection apparatus or fire suppression equipment, the separation of residents who can evacuate on their own from residents with reduced mobility, and designating refuge spaces. Several of these measures can prevent a nursing home from catching fire as well as extend the safe evacuation time of residents.

Some buildings now use fire resistant materials and smoke control components in order to establish fire safety compartments inside interior spaces. Based on the different performance-based designs of such compartment regions, these can be defined as fire compartments, smoke compartments, and refuge spaces [5]. A fire compartment is generally built with heat-resistant materials and designed to slow the burning rate and prevent a fire from entering a fully developed state. Smoke compartments are built with smoke control materials and prevent smoke or toxic gas leaking out when a fire occurs. Refuge spaces are based on a combination of fire and smoke compartments, and use materials with heat resistance and smoke control properties to keep the advantages of both compartments. To ensure the security of escape routes and convenience of operation, the openings of refuge safety compartments are always made of light materials that are easy to use, and thus objects such as fire retardant curtains have become a necessary part of nursing home design.

A fire retardant curtain is a moving fire retardant barrier that is usually rolled up in a roller box and which, if a fire occurs, can drop down to prevent a fire spreading in an interior space. Because they are safe and unobtrusive, fire retardant curtains are increasingly used in industrial and commercial buildings. Various heat resistance methods are applied in these curtains such as heat-resistant paint and multi-layer non-combustible fibers [6]. However, the cost of a drop-down curtain plus a water film cooling system is expected to be less than the cost of providing a fully rated fire separation. On the other hand, water spray cooling has received much attention in recent years for its ability to protect structures from heat.

### 1.3. The Purpose of This Study

Applying the water system to enhance the fire resistance of a glass pane or roller shutter has been demonstrated in many works [7–15]. Although the characteristics of a water film on a water absorbent curtain are different from those on a glass pane or roller shutter, the experimental and design experience obtained from the latter can be transferred to the former, because the basic mechanism of heat removal is the same. In the present study, we designed a water film cooling system to enhance the heat resistance and smoke resistance of a fire retardant curtain without a fire resistance rating, and so further examined the performance of the water film on the fire retardant curtain. Furthermore, a numerical evaluation was carried out in this study to develop a horizontal evacuation waiting space, using the proposed curtain system in an existing nursing home. The purpose of this study was to design a one-hour at least heat-resistance-rated and smoke resistance curtain with a water film system to apply in new or existing nursing homes to enhance the fire safety performance of such a building.

## 2. Materials and Methods

### 2.1. The Purpose of This Study

We investigated the heat and smoke resistance performance of the non-heat-resistant fire retardant curtain incorporating a water film using full-scale door/wall refractory furnace and smoke-barrier tests (Figure 1a). The furnace test provided data on the standard temperature heating curve for heat resistance. The construction and function of the furnace complied with ISO 834-1 [16], CNS 14803 [17], and ISO 3008 [18] for fire tests of fireproof doors (wood doors, steel roller shutters, etc.). The inner and outer dimensions of the furnace

were 430 (W) × 450 (H) × 100 (L) cm and 500 (W) × 520 (H) × l40 (L) cm, respectively. The dimensions of the available testing area, hereto referred to as the area directly exposed to the furnace fire, were 400 (W) × 400 (H) cm; the maximum testing duration was four hours.

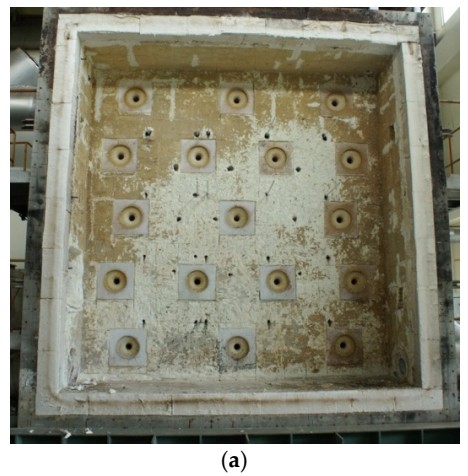

(a)

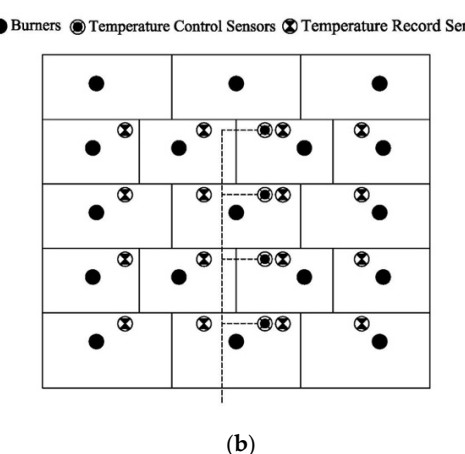

(b)

**Figure 1.** (**a**) Full-scale door/wall refractory furnace; (**b**) positions of thermocouples inside the furnace.

The steel structures used on the exterior of the furnace consist of the testing frame, air supply device, exhaust duct, and hood. Fireproof ceramic fibers capable of enduring temperatures of over 1400 °C were applied to the surfaces inside the furnace. The burning system contained flat-flame liquefied petroleum gas (LPG) burners, with ultraviolet (UV) flame sensors monitoring the flame state. The burner generated a flat yellow diffusion flame, and thus radiation and convection were the primary and secondary modes of heat transfer inside the furnace, respectively. Figure 1b displays the distribution of the burners and thermocouples in the furnace. Article 14803 of the Chinese National Standards (CNS) gives the relevant specifications for the mounting position of a thermocouple on a fire retardant curtain [17].

We installed a 300 × 300 cm curtain, with a thickness of 0.8 mm, on a testing frame (shown in Figure 2a). Figure 2b displays the distribution of thermocouples on the unexposed surface of the curtain. The fire retardant curtain specimens could experience pressure changes inside the furnace during the experiments, which would cause shaking of the curtain surface. For stabilization, spring devices were installed to secure the thermocouples with the curtain (Figure 2c).

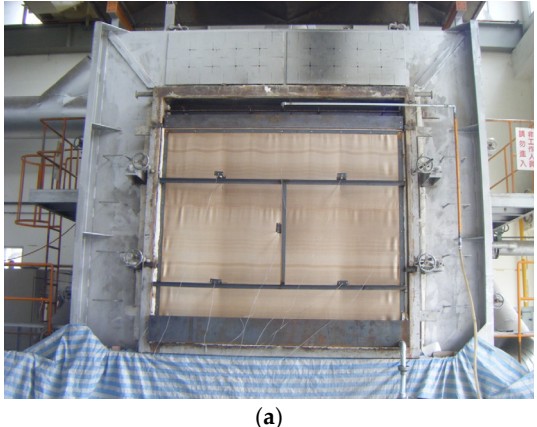

(a)

**Figure 2.** *Cont.*

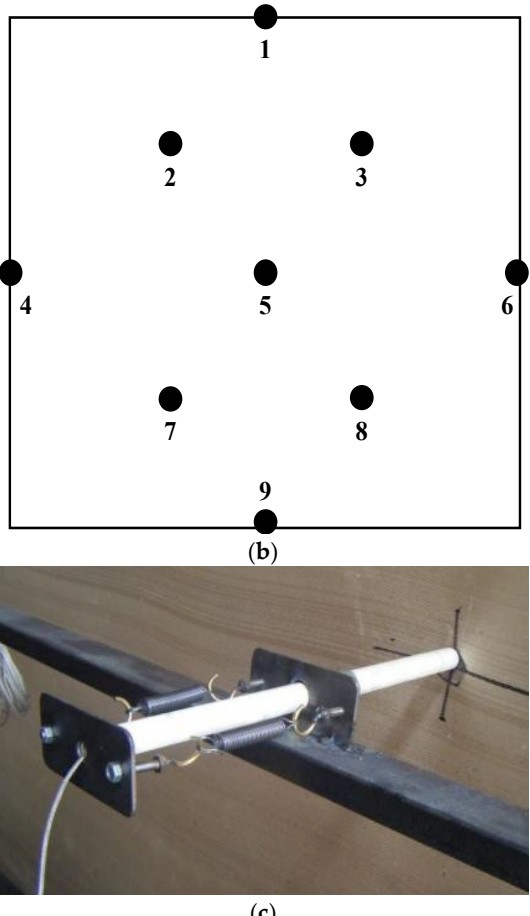

**Figure 2.** Thermocouples on the unexposed surface of the curtain: (**a**) installed on the testing frame, (**b**) positions of the thermocouples, (**c**) fixed with a spring device.

Figure 3 shows a schematic representation of our water film system using a hydraulic pump, steel pipe, and a piping system. The resultant water film aims to cover the non-exposed-to-fire surface of the curtain. Previously published test results demonstrated a perforated pipe that allowed water to exit through small holes along the length of the pipe to form a water film that covered the unexposed surface of the curtain [19,20]. In accordance with the calculation results [21], the first 28 holes from the end hole were 4 mm in diameter and the others were 4.1 mm. We placed the water inlet in the middle of the perforated pipe so that the jet velocity difference between the water inlet and the end hole was reduced. To match the specified flow rate, we also used a Grundfos single-phase 220 V–2210 W pump with a maximum head and flow rate of 52 m and 166 L/min, respectively. The relationship between the total water flow rate and uniformity of water film before the fire test is shown in Table 2. This result indicates that the minimum water flow rate needed to form a uniform water film is about 110 L/min (12.3 L/min/m$^2$).

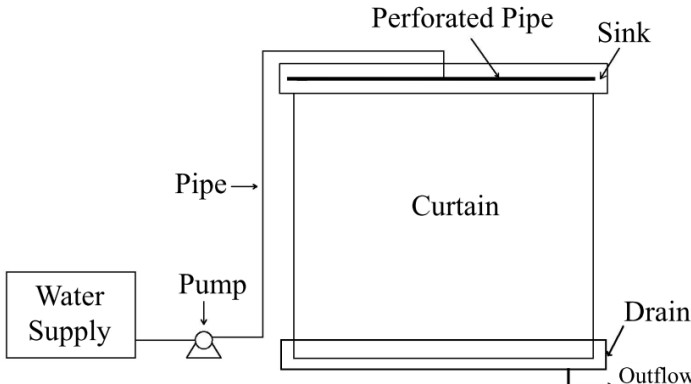

**Figure 3.** Schematic of the water film system.

**Table 2.** The relationship between the total water flow rate and uniformity of the water film before the fire test.

| Flow Rate at Each Hole (L/min) | Total Water Flow Rate (L/min) | Water Film Downward Velocity (cm/s) | Average Thickness of the Water Film (mm) | Form a Uniform Water Film (Y/N) |
|---|---|---|---|---|
| 2.354 | 65.92 | 33.46 | 26.2 | N |
| 2.900 | 81.20 | 35.51 | 27.8 | N |
| 3.834 | 107.34 | 38.72 | 30.3 | Y |
| 4.123 | 115.44 | 39.93 | 31.2 | Y |
| 4.599 | 128.76 | 43.66 | 34.2 | Y |

We also conducted an unexposed surface fire test (Figure 4). An accumulator was included to produce a consistent thickness of the water film horizontally across the width of the test curtain. A trough was installed at the bottom of the curtain to collect the remaining water from the curtain surface to investigate the relation between the evaporation of the water film and the heating rate of the furnace during the fire test. The volumetric flow rate of the return water was measured by a calibrated container of 100 L capacity.

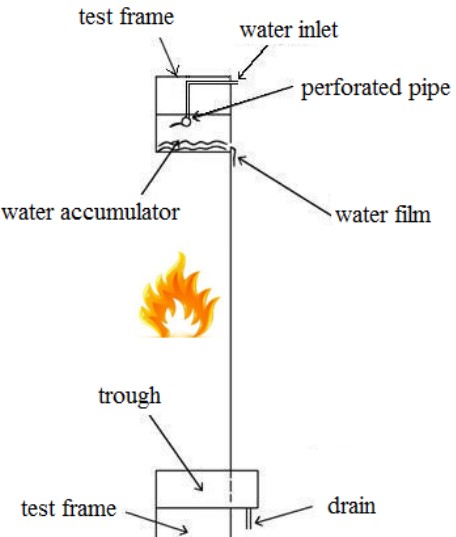

**Figure 4.** Schematic of the unexposed-surface fire test (R: unexposed surface/L: exposed surface).

In the fire tests, the objective of the heating duration was 90 min. Tests of the fire-proof curtain were conducted with and without water film in contrast. The operating pressure on

the perforated tube, Pop, was set to 196 kPa to maintain a constant depth in the accumulator so that a uniform film depth could be maintained over the curtain before the testing. The total flow rate was 111 L/min (12.3 L/min/m$^2$), the velocity of the water film vs was 30.3 cm/s, and the average thickness of the water film was 1.27 mm. All of the data signals were collected and converted using a data acquisition system (Yokogawa DA100). The data acquisition system then transmitted the converted signals through a DAQ interface to a personal computer for further operation. Additionally, a smoke-barrier system was used to investigate the smoke resistance of a curtain with a water film system in this study, as shown in Figure 5a. It consisted of a refractory wall/door testing furnace and a smoke-barrier testing device (Figure 5b). This smoke-barrier system could be used to simulate and examine the smoke behavior through openings (vents or holes), doors, and shutter assemblies in a building fire.

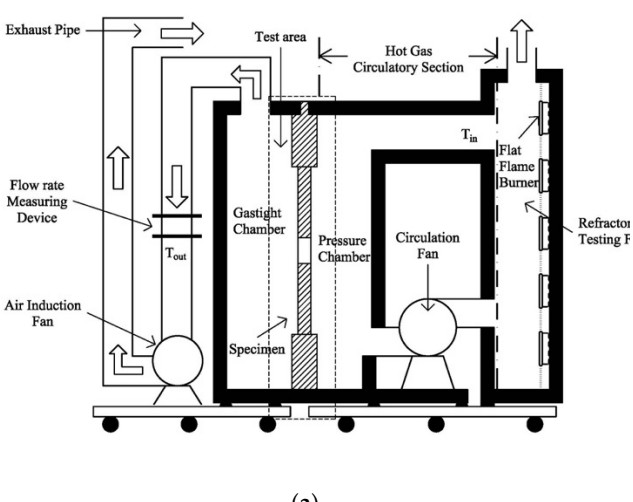

(**a**)

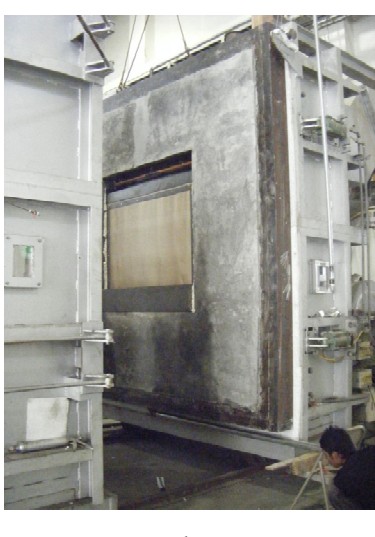

(**b**)

**Figure 5.** The schematic and images of the smoke-barrier system: (**a**) smoke-barrier system, and (**b**) smoke-barrier testing device.

In this study, the same type of curtain was used to examine the smoke leakage characteristics. The test surface area of the curtain was 105 cm (W) × 80 cm (H). The pressure difference was controlled at 10, 25, and 50 Pa across the specimen by changing the capacity of the air-induction fan. The temperature in the pressure chamber was controlled and maintained at room temperature and 200 °C based on ISO 5925-1 [22].

### 2.2. Evaluation of the Application of a Fireproof Curtain with the Water Film System in a Nursing Home

The flow and deposition of smoke for a fire on the sixth floor of a seven-floor complex nursing home was simulated using Fire Dynamics Simulator (FDS 6.5.3, NIST, Gaithersburg, MD, USA). This nursing home cannot be used due to failure to comply with the fire regulations. Thus, the use of a temporary evacuation relay station built using a fireproof curtain with a 1.5-h fire-resistance period developed in this study was also explored to see its effect on the safe evacuation of the residents.

FDS is a field-based fire simulation software developed by the Building and Fire Research Laboratory, National Institute of Standards and Technology (NIST). It is a CFD model, which takes the fluid motion in a fire as the main model object. The results of the calculations are processed by post-processing software (Smokeview 6.4.4, NIST, Gaithersburg, MD, USA), with which the plume, gas temperature distribution, and flow field distribution in the fire scene are displayed in animation, the fire scenario can be modeled rapidly, and the fire scene information can be known in time. Its open source nature makes it excellent for research. Meanwhile, the FDS simulation program is widely

adopted in the fire safety industry and its results are commonly acceptable [23,24]. Our review indicated that approximately half of existing published studies used the FDS software to design smoke ventilation systems, automatic fire suppression systems, and fire alarm systems, while the other half simulated the residential and industrial fire protection engineering design.

Figure 6a shows a three-dimensional physical model of the nursing home drawn using FDS. The simulated nursing home was 35 m in length, 36 m in width, and three meters in height, with its interior configured with 14 rooms, one medical nursing station, and one kitchen. The two green blocks are solid, inaccessible parts. This study set up a total of two fire points. Fire point in Figure 6b was above a bedside and the ignition source was a cigarette butt, and fire point B in Figure 6c was in the kitchen and the ignition source was a stove. The heat release rate of a fire was 1 MW. All partition walls of each room were set as a solid boundary, with gypsum board as the wall material. The bed in each room was set as a cubic solid. The exit door was set as an opening applied to the exterior boundary of the computational domain. The proposed curtain containing an additional water film was defined as mixtures of solid material within the same layer. The fireproof curtain began to activate after a fire had been burning for 30 s. The simulation time was 300 s. In terms of the grid size setting, the computational grid in FDS is Cartesian and cubic cells. Dependence of the simulated results by FDS on the grid resolution is a well-known fact [25,26]. The cell sizes (dx) could be determined using the characteristic fire diameter (D*) and cell size ratio (D*/dx) that should accurately resolve the fire simulation based on the heat release rate [25]. Based on the above-mentioned parameters, the characteristic fire diameter (D*) in this fire simulation was 0.959. A reference within the FDS User Guide [26] indicated that a D*/dx ratio between 4 and 16 could accurately resolve fires in various scenarios. Thus, the minimum computing grid was set to 0.175 m; the maximum grid size was 0.33 m, so the total grid had 400,000 (200 × 200 × 10) points.

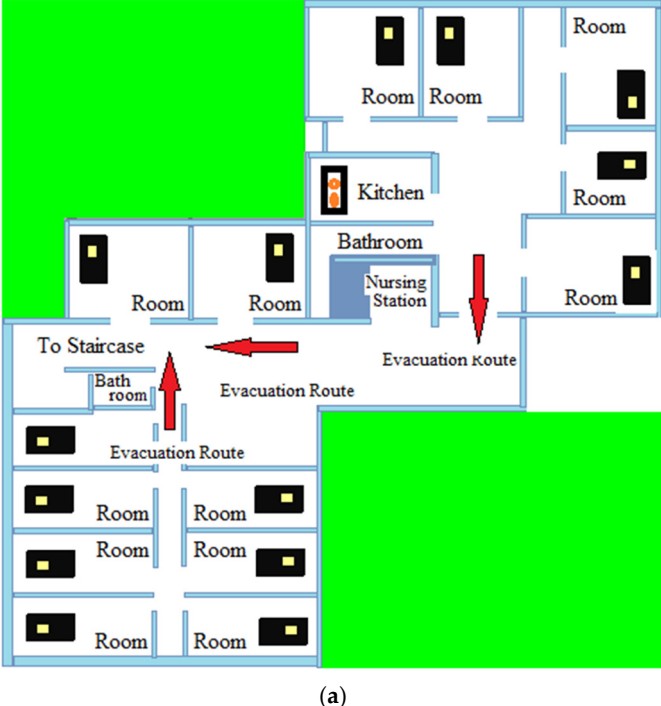

(a)

**Figure 6.** *Cont.*

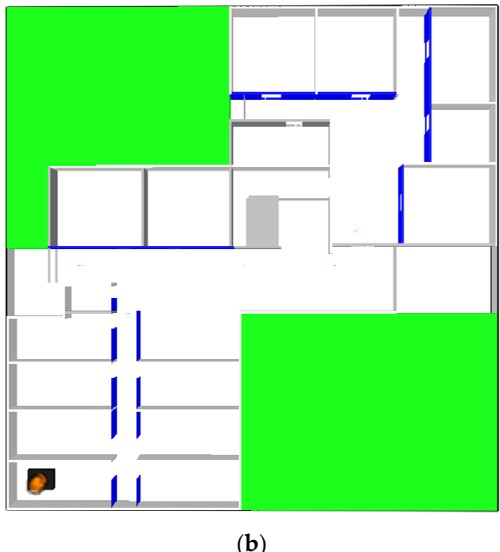
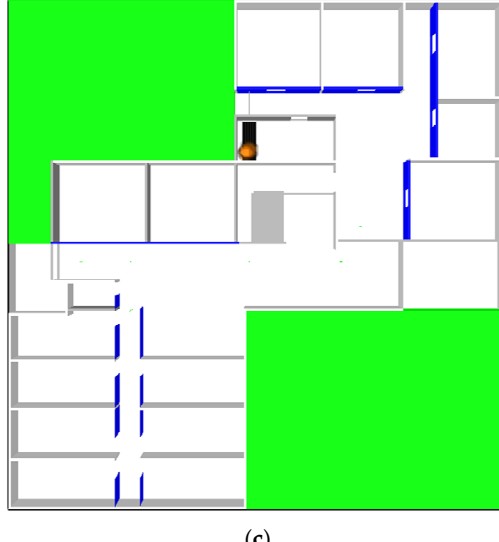

(**b**)  (**c**)

**Figure 6.** (**a**) Three-dimensional physical model and fire evacuation direction of the nursing home; (**b**) fire point above a bedside and the ignition source was a cigarette butt; (**c**) fire point in the kitchen and the ignition source was a stove.

## 3. Results and Discussions

In this study, a fire retardant curtain without/with a water film system was tested with the full-scale door/wall refractory furnace and the smoke-barrier system. The objective of this study was to enhance the fire resistance performance of our proposed curtain with a water film system to a fire rating of 90A. Since the fire and smoke-barrier test is quite expensive and time-consuming, we only conducted two tests under each condition to obtain the reproducibility of the experimental results. Finally, a refuge safety compartment built with the curtain in an existing nursing home was simulated to prolong the refuge period. Our results showed that smoke resistance was enhanced to match ISO 5925-1 [22]. A previous study tested and showed that a fire retardant curtain without a water film could only provide fire integrity for 30 min and fire resistance for less than five minutes [21].

### 3.1. Heat Resistance Experiment for the Fire Retardant Curtain without a Water Film System

Figure 7 shows the temperature variation of each thermocouple on the unexposed surface of the curtain and the average temperature inside the furnace during the fire test. Our results show that the fireproof curtain was uniformly heated, as there was limited temperature variation. The average fireproof curtain could bear up against fire and flame, but in the absence of a water film system, the surface temperature increased as fire continued to burn. The temperature on the unexposed surface basically followed the furnace temperature with a difference around 200 °C. The LPG fuel consumption in this 30 min fire test was 10.86 L. Thus, a typical fireproof curtain is prone to heat-induced degradation, which led to the breakdown of the curtain occurring at 30 min after ignition (as shown in Figure 8). Our outcome demonstrated that fireproof curtains without a water film system only had fire integrity up to 30 min; heat resistance was approximately five minutes.

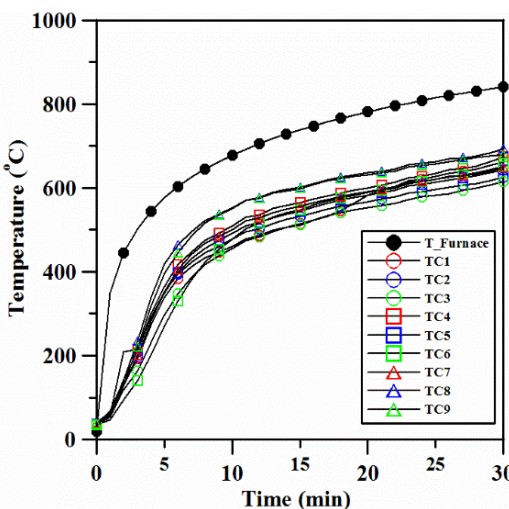

**Figure 7.** The temperature distribution on the unexposed surface of the fireproof curtain without a water film.

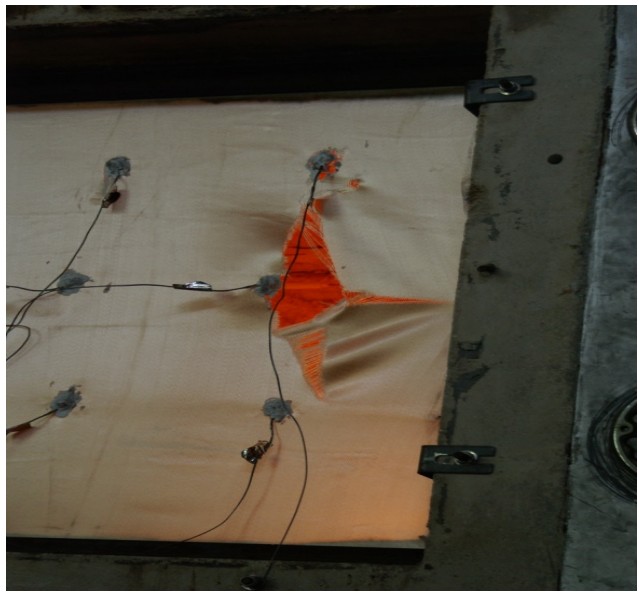

**Figure 8.** The breakdown of the fireproof curtain without a water film.

### 3.2. Heat Resistance Experiment the Fire Retardant Curtain with a Water Film System

The experimental record is summarized in Table 3. Each experimental stage and photos of important observations are shown in Figure 9. As illustrated, the amount of water film being heated into water vapor during the experiment was minimal; the amount of water supplied in this experiment was sufficient to form a water film on the curtain's surface. Since the surface was uneven, water films generated dry zones on the curtain, which ultimately led to the generation of hot spots. After the hot spots had been continuously heated, a burnt hole appeared at 57 min. Another hot spot zone and burnt hole appeared after 90 min into the experiment. However, direct fire transference was not observed when a standard cotton cloth was placed in the burnt holes, indicating that the curtain still had heat-resistant properties. The LPG fuel consumption at 30 and 90 min during the fire testing of the fire retardant curtain with a water film system was 15.56 and 56.28 L, respectively. Because the water film on the unexposed surface of the curtain could absorb some of the heat in the furnace, the LPG fuel consumption in a case of fire testing of the fire retardant curtain with a water film system was much greater than that without a water film system.

**Table 3.** The experimental record.

| Time | Description of Phenomena | Time | Description of Phenomena |
|---|---|---|---|
| 0:00 | Water film began to flow down. Uneven water film distribution. | 0:57:19 | The first hole appeared on the curtain. |
| 0:01 | Door/wall furnace ignition. | 1:16:18 | Leakage was measured. The curtain was unburned. |
| 0:23:46 | A hot spot began to appear on the curtain. | 1:30:00 | A second hole appeared on the curtain. |
| 0:43:44 | The second hot spot began to appear. | After 1:30:00 | End of the experiment. |
| 0:52:00 | Another two hot spots appeared on the curtain. | | |

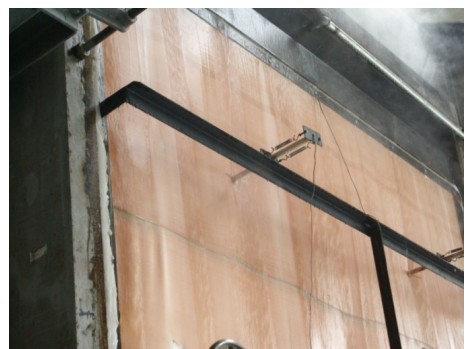

Water film began to flow down. Uneven water film distribution. (0:00:00)

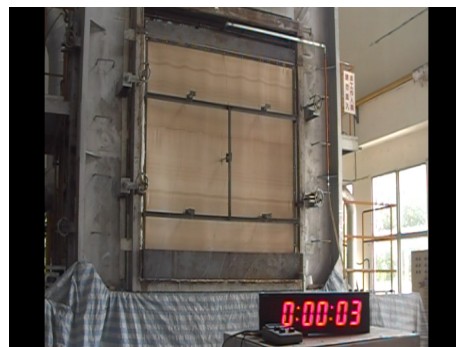

Door/wall furnace ignition. (0:00:03)

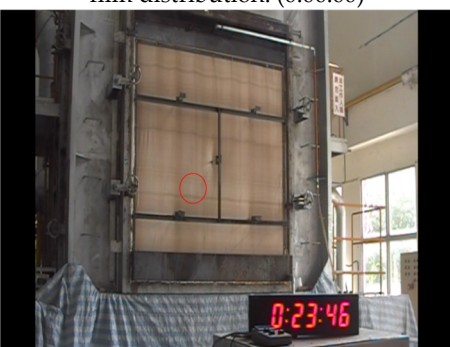

A hot spot began to appear on the curtain. (0:23:46)

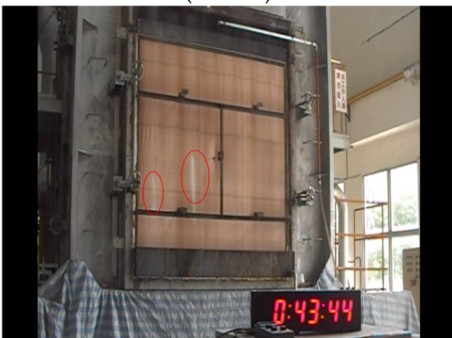

The second hot spot began to appear. (0:43:44)

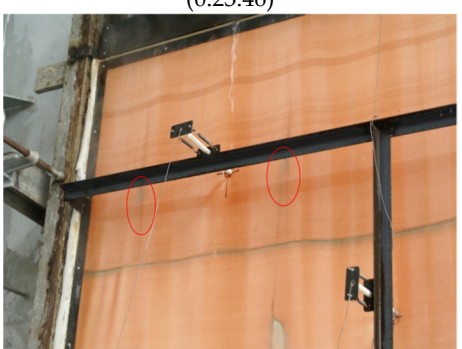

Another two hot spots appeared on the curtain. (0:52:00)

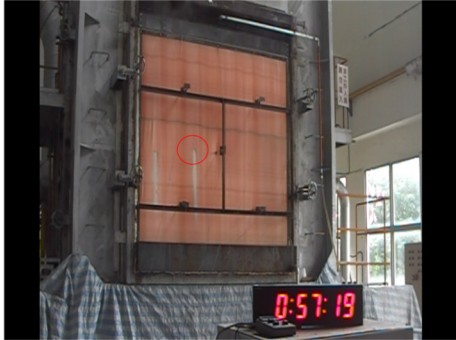

The first hole appeared on the curtain. (0:57:19)

**Figure 9.** *Cont.*

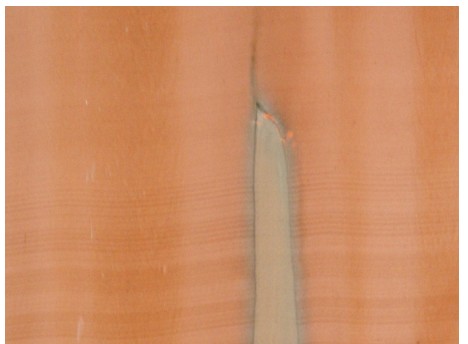

The first hole appeared on the curtain.
(0:57:19)

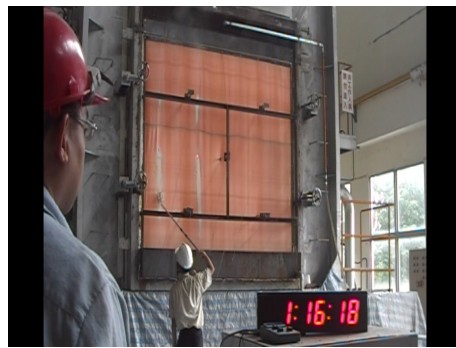

Leakage was measured. The curtain was un-
burned. (1:16:18)

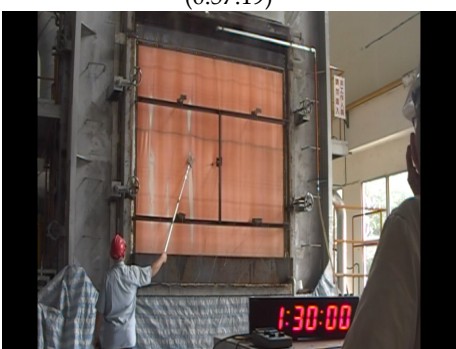

A second hole appeared on the curtain.
(1:30:00)

**Figure 9.** The experimental photos of the important observations.

Figure 10a–d shows the heating curves for all points inside the wall/door furnace. The experimental results showed that all of the temperatures were within the heating curve range specified in CNS 14803 [17]; however, the heating rates at each position were not consistent, so the heating on the curtain was uneven. The combustion and pressurization processes in the furnace expanded and bent the curtain externally.

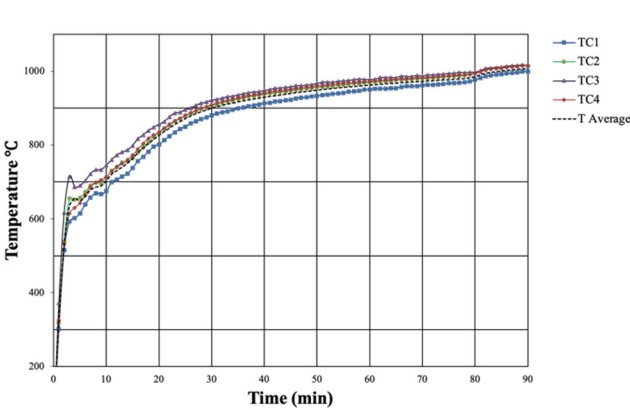

(**a**) TC 1~TC 4 inside the furnace

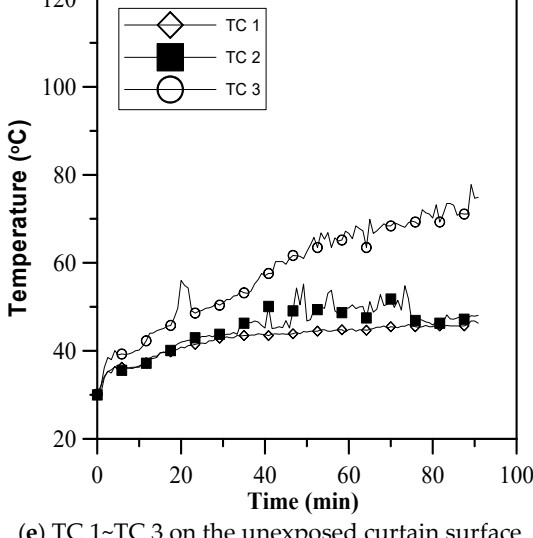

(**e**) TC 1~TC 3 on the unexposed curtain surface

**Figure 10.** *Cont.*

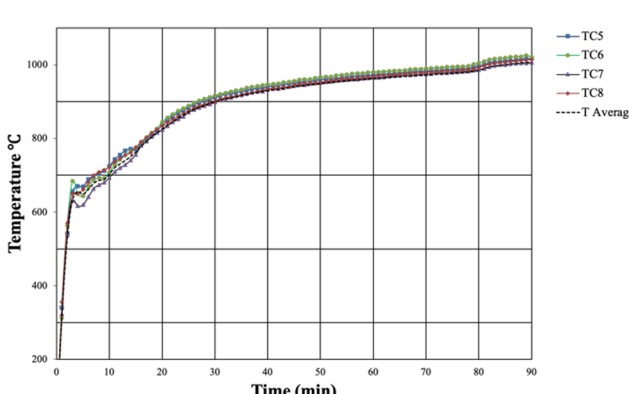

(**b**) TC 5~TC 8 inside the furnace

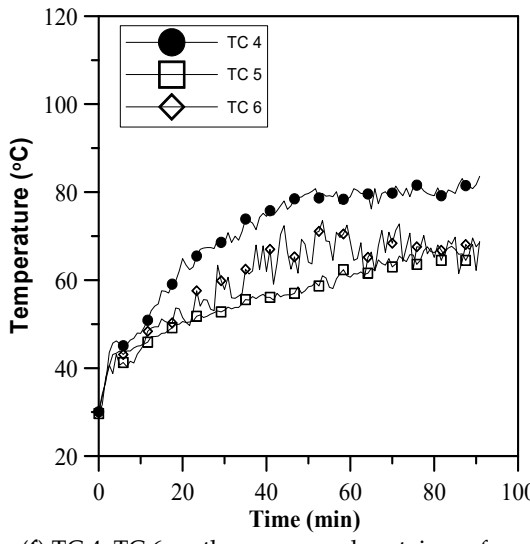

(**f**) TC 4~TC 6 on the unexposed curtain surface

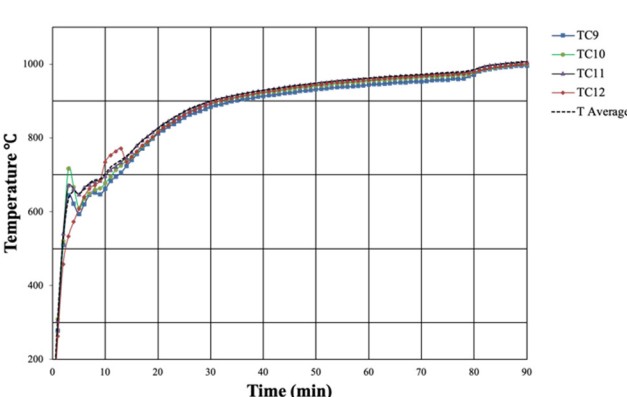

(**c**) TC 9~TC 12 inside the furnace

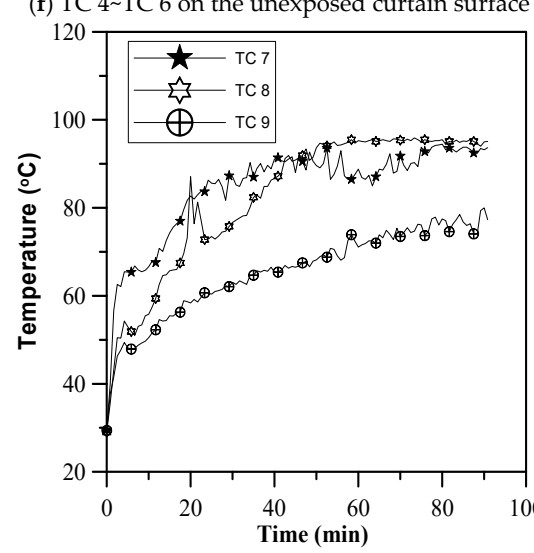

(**g**) TC 7~TC 9 on the unexposed curtain surface

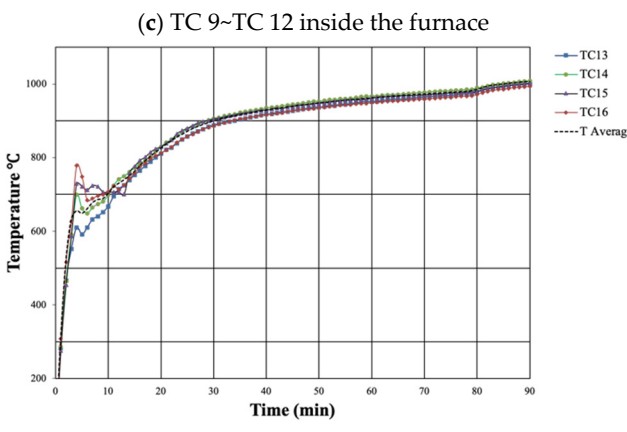

(**d**) TC 13~TC 16 inside the furnace

**Figure 10.** Temperature curves of the measurement points; (**a**) 1–4, (**b**) 5–8, (**c**) 9–12, and (**d**) 13–16 inside the wall/door furnaces; (**e**) 1–3, (**f**) 4–6, (**g**) 7–9 on the unexposed curtain surface.

Figure 10e–g shows the heating curves at each measurement point at the top, central, and bottom part of the curtain. Of all the measurement points, point 1 was the closest to the water film output, where the water film only absorbed the minimum heat, so the highest temperature achieved at this position in the experiment was only 45 °C. TC 2 and

TC 3 were both similarly covered by the water film, but there were significant differences in the amounts of water covering these two points, and therefore the heating rate of point 3 was greater than that of point 2. At TC 5, since there was water covering the curtain, the highest temperature reached in the experiment was only 60 °C. TC 4 and TC 6 were similarly covered by a water film, so the temperatures did not exceed 100 °C. However, the temperature difference between these two points could be because the water film was not even or did not have a symmetrical covering, in addition to the fact that the furnace temperature was not uniform, causing the temperature of TC 4 to be higher than the temperature of TC 6. TC 9 was located at the very bottom of the water curtain, where the temperature was the lowest among the three temperatures. The measured temperature at TC 9 indicated the average temperature of the return water in the drain. TC 7 and TC 8 had similar water film coverings, but the downward flowing water film absorbed the heat on the curtain, causing the temperatures to be close to 100 °C.

Figure 11 shows a plot of the temperature distribution on the curtain surface measured using an infrared thermal imager. The measurement points 1–5 denoted in the infrared thermal imager correspond to the measurement points 2, 3, 5, 7, and 8 of the thermocouples. As demonstrated, the right-side temperatures of the curtain were lower because of the non-uniformly formed fire retardant surface on the curtain. However, after the experiment had continued for 90 min, the temperature difference on the thermal imager was not great, suggesting that the temperature change in the curtain tended to be stable, while the dry zone gradually expanded over time. For the duration of the experiment (90 min), apart from the two hot spots that began to appear on the left side of the curtain at 44 min, the surface temperatures were all less than 150 °C. For the remaining zones covered by the water film, the surface temperatures of the curtain were all below 100 °C.

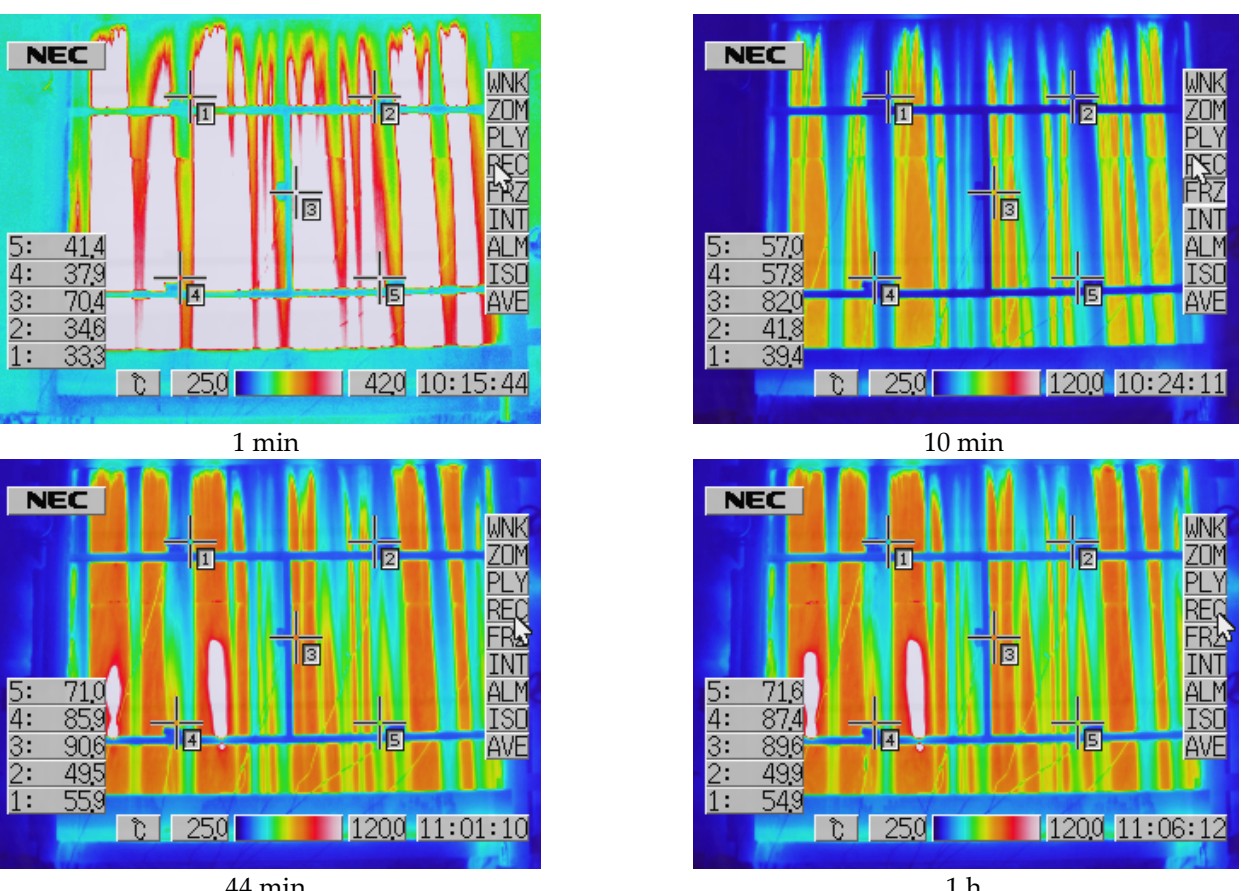

**Figure 11.** *Cont.*

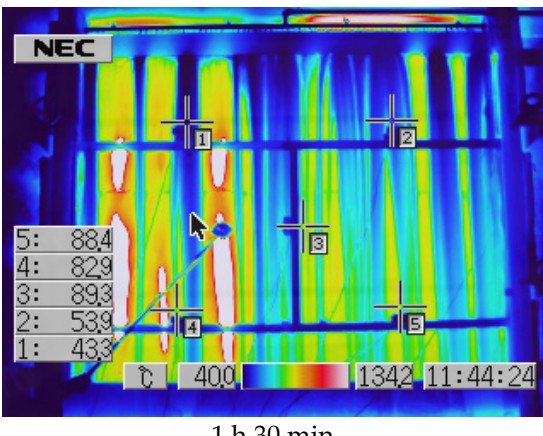

1 h 30 min

**Figure 11.** The temperature distribution on the curtain surface measured with an infrared thermal imager.

Our experiment results demonstrated that low uniformity of the water film led to more water being required and high temperature areas on the surface of the curtain. The dimensions of the curtain dominated the scale and difficulty of the water film system. Increasing the width and height of the curtain would require more water and may induce potential non-uniformity. When the water flowed downward through the heated curtain, the surface temperature increased (as shown in Figures 10 and 11). Notably, breakdown may occur at the lower part of curtain with insufficient water supply. In addition, the water supply is the most important part of the system. In this study, the water supply was set at the minimum needed to form a uniform water film. The amount of water is determined by the two parameters above-mentioned, and the supplying device should avoid intrinsic non-uniformity of the water film. The surface temperature can be suppressed to below 100 °C with a stable and uniform water supply.

*3.3. Smoke Control Experiment*

Figure 12 shows the smoke control performances of (a) the fire retardant curtain without a water film system at normal temperature; (b) the fire retardant curtain with a water film system at normal temperature; and (c) the fire retardant curtain with the water film system at 200 °C. In Figure 12, q represents the amount of smoke leakage from the specimen, q_fail is the criteria rate of the leakage of ambient (cold) and medium (warm) temperature smoke from one side of the specimen to the outside. At normal temperatures, the smoke control performances of the fire retardant curtain were 0.08 m$^3$/min (10 Pa), 0.18 m$^3$/min (25 Pa), and 0.56 m$^3$/min (50 Pa). At 50 Pa, the smoke control performance of the fire retardant curtain (0.56 m$^3$/min) exceeded the standard value (q_fail) of 0.42 m$^3$/min. The fire retardant curtain thus cannot meet the standards of ISO 5925-1 [22]. For the fire retardant curtain with the water film under normal temperature conditions, the smoke control performances were 0.08 m$^3$/min (10 Pa), 0.17 m$^3$/min (25 Pa), and 0.23 m$^3$/min (50 Pa). At 200 °C, the smoke control performances of the fire retardant curtain with the water film system were 0.02 m$^3$/min (10 Pa), 0.08 m$^3$/min (25 Pa), and 0.16 m$^3$/min (50 Pa). Therefore, when the water film system was applied in the curtain, the smoke control performance was enhanced and under normal and intermediate temperature conditions, it could meet the standards of ISO 5925-1 [22].

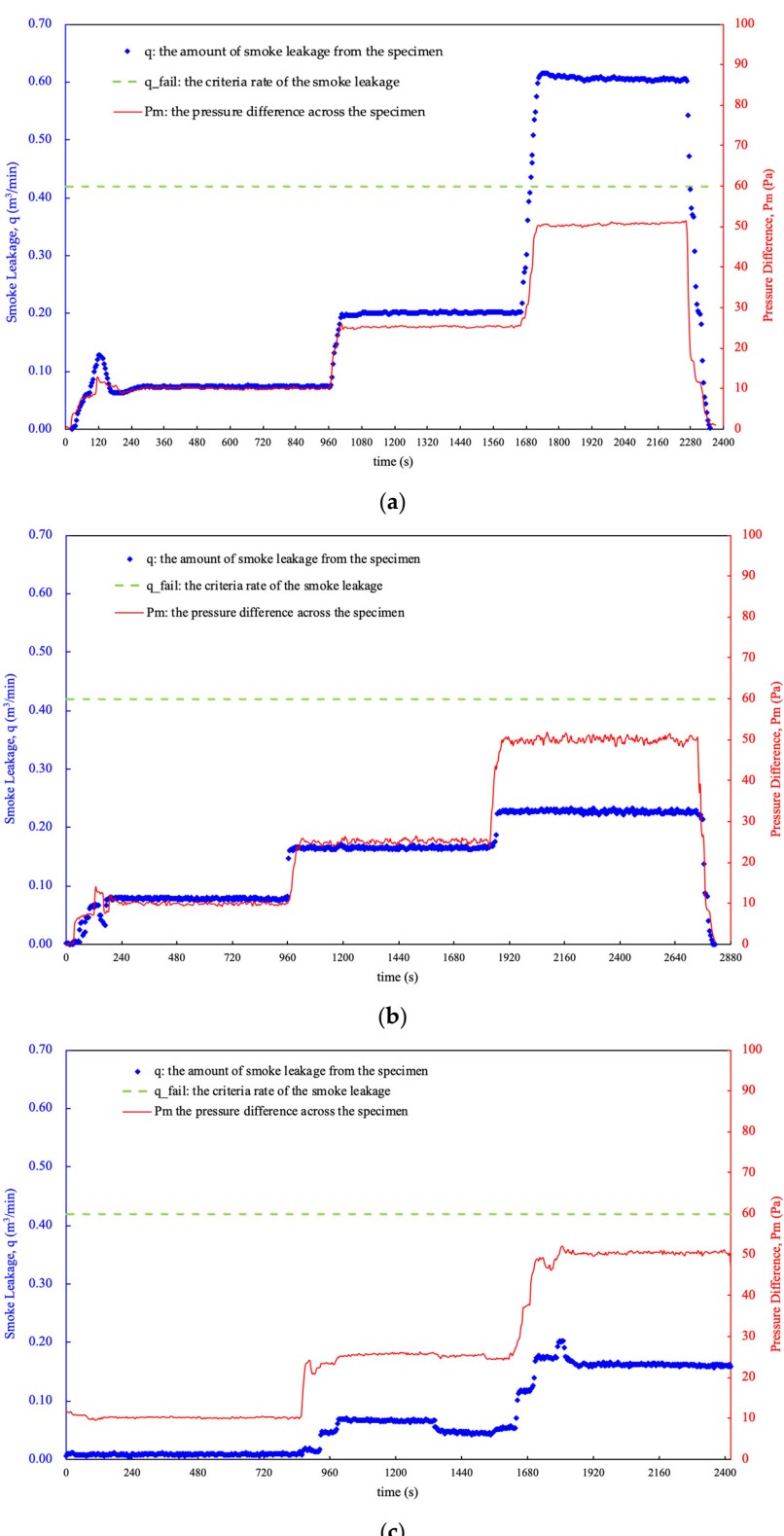

**Figure 12.** Smoke control performances of the fire retardant curtain: (**a**) without a water film system at normal temperature, (**b**) with a water film system at normal temperature, and (**c**) with the water film system at 200 °C.

### 3.4. Simulation Results of the Application of a Fireproof Curtain with the Water Film System in a Nursing Home

The simulation results for the selected nursing home with only a detector and sprinkler set are shown in Figure 13. Figure 13a shows the current fire evacuation directions for the nursing home. Each floor of the nursing home only had one escape exit, connecting straight to the stairs for evacuation and escape. This research assumed that the doors of all of the rooms were open, so 30 s after the fire at the fire point above a bedside started burning, thick smoke diffused into three rooms. Thick smoke spread to the middle aisle after approximately 60 s. After 120 s, the thick smoke spread to the rooms in another zone. When a fire occurred at a fire point in the kitchen, thick smoke diffused into four rooms after burning for 30 s. After 60 s, thick smoke spread to the middle of the aisle. After 120 s, the thick smoke spread to the escape exit. According to these results, when the fire is burning, the smoke diffusion area is directly proportional to the burning time. If a fire is not immediately detected and extinguished in the initial stage, then the smoke diffusion speed will shorten the escape time for disadvantaged evacuees in a nursing home. When other fire-fighting measures are not taken, there will be even greater casualties.

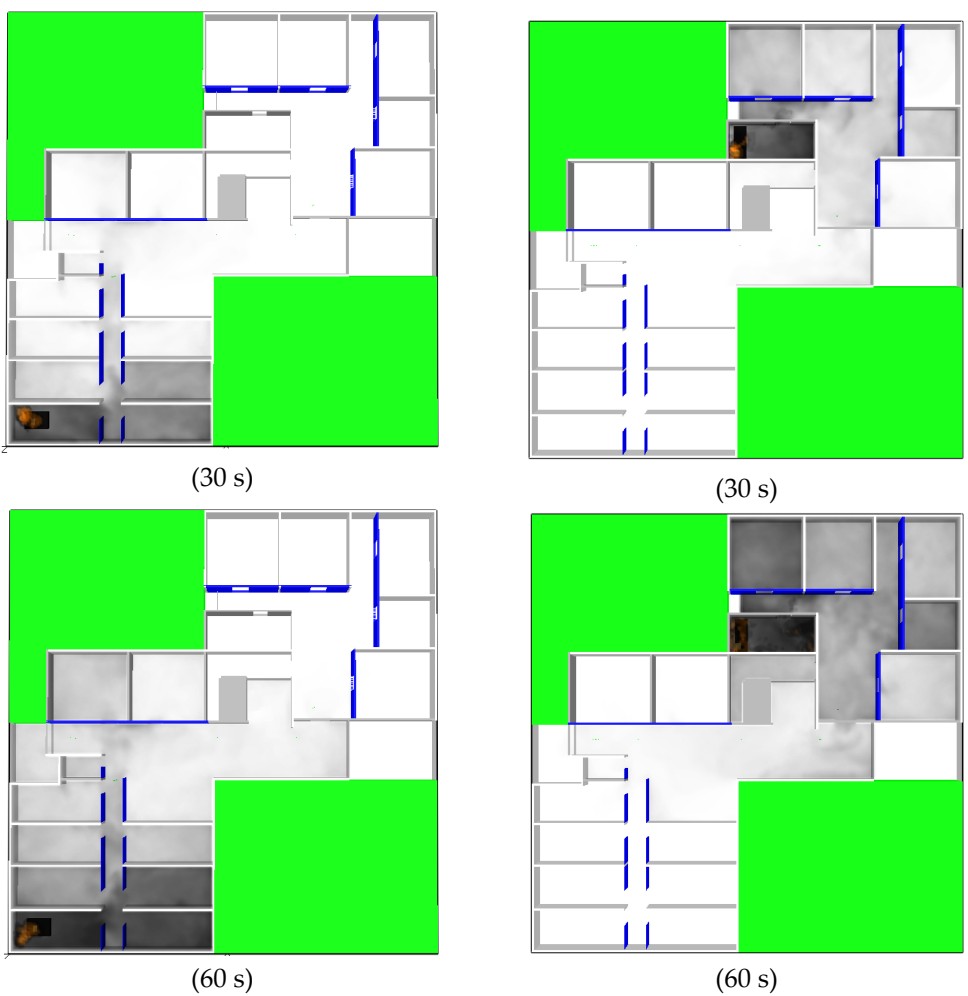

(30 s)

(30 s)

(60 s)

(60 s)

**Figure 13.** *Cont.*

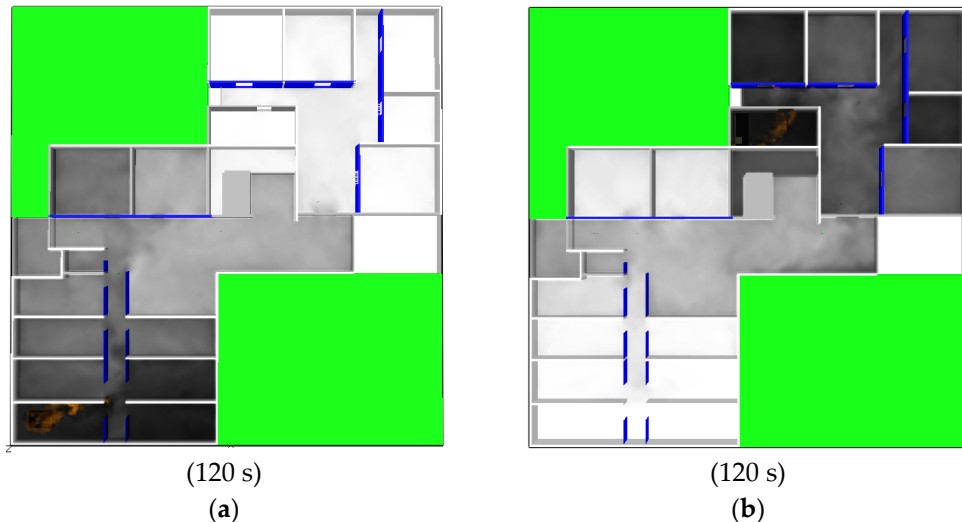

(120 s)  (120 s)

(**a**)  (**b**)

**Figure 13.** Smoke spread simulation in a nursing home: (**a**) fire point above a bedside, and (**b**) fire point in the kitchen.

Figure 14 shows a schematic diagram of the temporary evacuation relay station built using the fireproof curtain with the 90-min fire-resistance period. The simulation results for fires occurring in the former two locations are shown in Figure 15. As indicated by these results, the fireproof curtain began to activate after a fire had occurred for 30 s, regardless of whether the fire occurred at a fire point above a bedside or in the kitchen. The system can thus effectively block smoke diffusion to another zone. As indicated by the results from the two cases above, the temporary evacuation relay station can not only block the diffusion of smoke, but also be a safe evacuation waiting zone for disadvantaged evacuees in a nursing home.

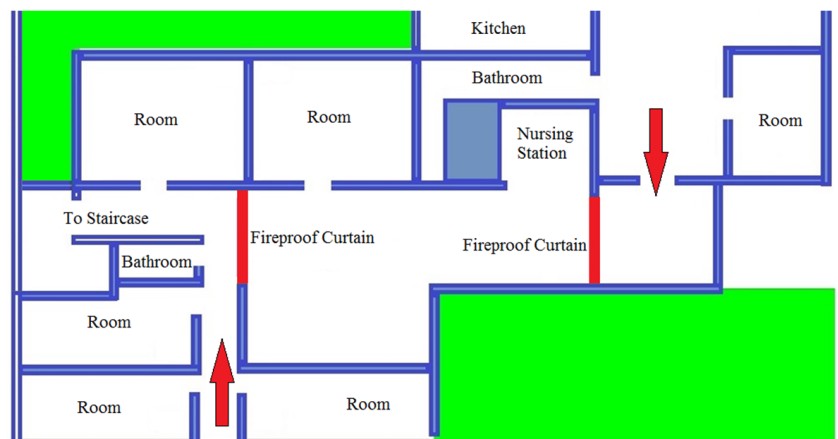

**Figure 14.** Schematic diagram of the temporary evacuation relay station built using a fireproof curtain with the 1.5-h fire-resistance period.

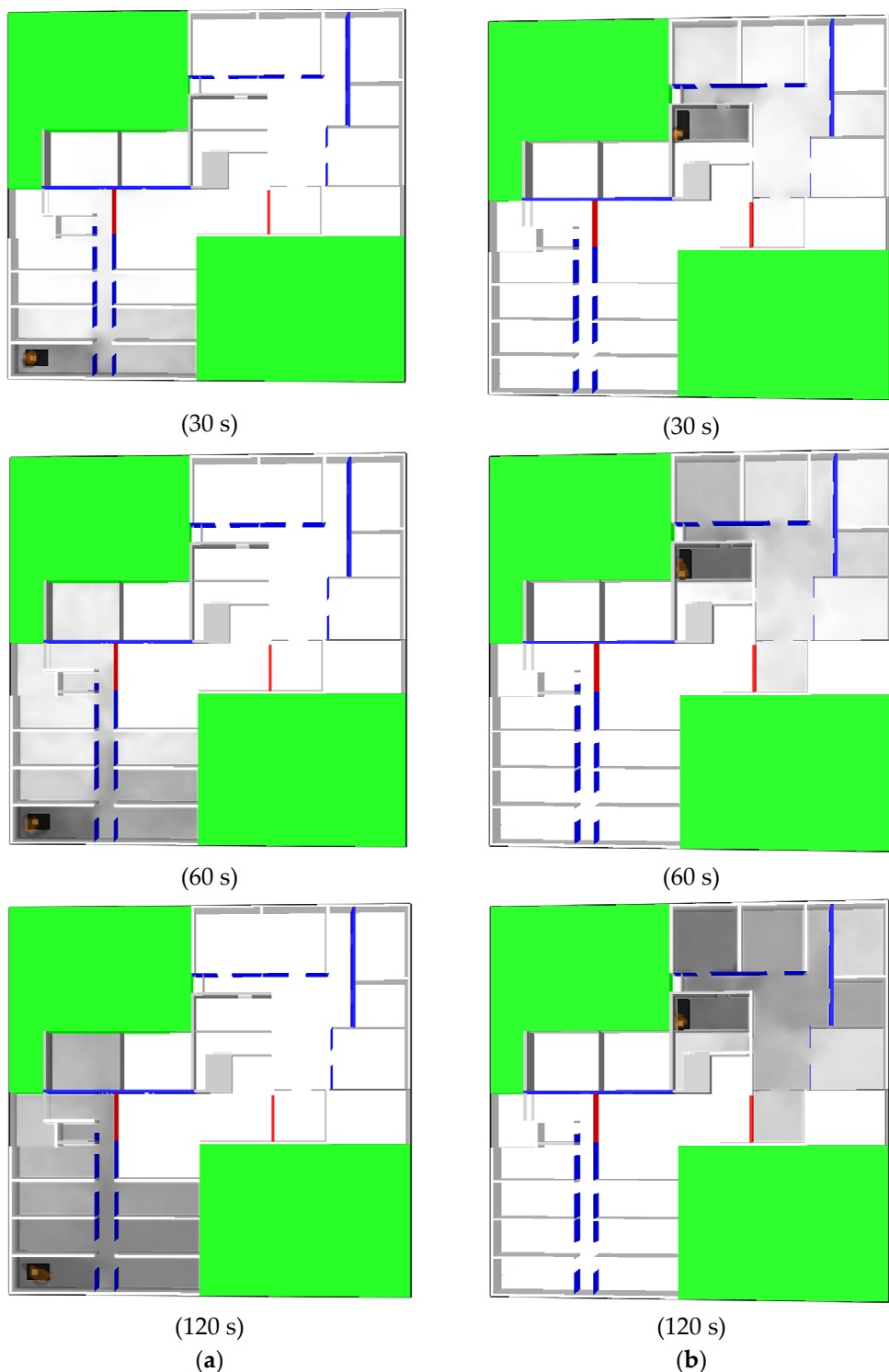

**Figure 15.** Smoke spread simulation for a temporary evacuation relay station built with a fireproof curtain with the 1.5-h fire-resistance period: (**a**) fire point above a bedside, and (**b**) fire point in the kitchen.

The design guide for the application of this proposed fireproof curtain with a water film system in the nursing home could be described as follows:

1.  A fireproof curtain system should be installed to provide a horizontal evacuation waiting space for occupants in a nursing home that is reasonably safe from fire and the products of combustion.

2.　　The fireproof curtain system with a water film system should be connected to the fire detector system. The fireproof curtain should be activated from the fire detector system, and the water supply to form the water film on the surface of the fireproof curtain is then activated after the fireproof curtain is rolled down completely.

3.　　The building should ensure adequate access to a water supply. The water storage tanks should be provided with high- and low-level water switches that are connected to the site fire alarm system and initiate supervisory alarm signals.

## 4. Conclusions

This study examined the heat resistance properties and smoke control performance of a fire retardant curtain with a water film system by using a full-scale 3 m × 3 m door/wall refractory furnace and smoke-barrier testing system. Our experimental results indicate that when a sufficient amount of water is supplied to form a water film on the curtain surface, the proposed fire retardant curtain with a water film system in this study had heat resistance properties for 90 min (90A). However, the curtain surface does not have a smooth face, so an uneven water film generates dry zones on the curtain, which would create hot spots. If the issue of uneven water outflow can be improved or the amount of water outflow can be increased, then the heat resistance properties of the fire retardant curtain can be enhanced. The application of the water film system to the curtain enhanced the smoke control performance in both the normal and intermediate temperature conditions, thus achieving the standards of ISO 5925-1 [22]. In addition, a temporary evacuation relay station using the proposed fireproof curtain was then modeled with FDS 6.0 in the space of an existing nursing home to explore its effectiveness in improving evacuations. The simulated results indicated that the fireproof curtain began to activate after a fire had been burning for 30 s and this system can thus effectively block smoke diffusion to another zone and can be used as a safe evacuation waiting zone, reducing the casualties due to fire in a nursing home.

**Author Contributions:** Conceptualization, T.-H.L., S.-K.L., and C.-Y.C.; Methodology, S.-K.L.; Software, C.W.; Validation, S.-K.L. and C.-Y.C.; Formal analysis, S.-H.C.; Investigation, S.-H.C.; Data curation, S.-H.C.; Writing—original draft preparation, S.-H.C.; Writing—review and editing, C.W. and C.-Y.C.; Visualization, S.-H.C.; Supervision, T.-H.L.; Project administration, S.-K.L. All authors have read and agreed to the published version of the manuscript.

**Funding:** This research received no external funding.

**Data Availability Statement:** Not applicable.

**Acknowledgments:** The authors wish to thank the Architecture and Building Research Institute, Ministry of Interior, Taiwan for the financial support.

**Conflicts of Interest:** The authors declare that there are no conflicts of interest regarding the publication of this paper.

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
