# Peer review of "Improvement of Fire Safety Performance for Nursing Homes by Using Fireproof Curtains with a Water Film System"

_buildings, doi:10.3390/buildings12101590_

Round 1
Reviewer 1 Report
This article is very practical and has innovative aspects, and it is written properly and is related to the field of the BUILDING JOURNAL, and it is written without problems. I suggest that the article be published in this JOURNAL.
Author Response
Many thanks to the reviewer for the acknowledgment.Reviewer 2 Report
The scientific quality of the article is good and the article contains the information needed to explain the essence of the problem and its solution.
The article is relevant to the journal Buildings.
Reviewer comments:
Introduction. The introduction is sufficient.
The experimental specification section describes experimental methods. Sections sufficiently describes the topic of article. Figures (in all article) have good resolution except Fig.5. Please provide fig. 5 in higher resolution and size.
The results and experiments are sufficiently explained in the Results section. Please consider y-axis resolution in some figures (e.g.7, 8). Please modify fig. 9 y-axis. Graphs should not have common points on the y-axis.
The summary and conclusion section correctly and sufficiently explain the measurement results.
Notes:
1. Please put in the article an estimate of heat production during combustion in the burner. Is there any amount of heat production that will damage the curtains?
Author Response
Point-by-point reply to reviewers’ comments
REVIEWER 2 comments:
1. The introduction is sufficient.
RESPONSE: Many thanks to the reviewer for the acknowledgment.
2. The experimental specification section describes experimental methods. Sections sufficiently describe the topic of article. Figures (in all article) have good resolution except Fig.5. Please provide fig. 5 in higher resolution and size.
RESPONSE: We are very grateful to the reviewer for the valuable suggestion. We offered the higher resolution photo and illustration in Fig.5.
3. The results and experiments are sufficiently explained in the Results section. Please consider y-axis resolution in some figures (e.g.7, 8). Please modify fig. 9 y-axis. Graphs should not have common points on the y-axis.
RESPONSE: We are very grateful to the reviewer for the valuable suggestion. Figure 8 in the original manuscript has been removed. We also combined Figure 7 and Figure 9 into a new figure, for the convenience of the reader to compare the temperature variation in the furnace and on the unexposed surface of the curtain over time. Please refer to Figure 10 in the revised manuscript.
4. Please put in the article an estimate of heat production during combustion in the burner. Is there any amount of heat production that will damage the curtains?
RESPONSE: We are grateful to the reviewer for critically reviewing our manuscript and providing valuable comments. The total LPG energy consumption for the fire testing of the fire retardant curtain without/with a water film system has been added in the revised manuscript. Please refer to Line 249 and Line 269~274.
Reviewer 3 Report
Dear Authors
Thank you for your work. I think it is well structured and easy to understand. However I think it should be more ambitious in order to have real scientific value. For example, you do not present the results of the tests without water in the same way as the tests with water, which limits the comparison. The statistical value is compromised when they do only one test without repetitions. Furthermore, they analyse one scenario, without explaining why, when it would be much more valuable if they varied the conditions (for example, the thickness of the film, which as they say is so important).
I leave below more suggestions for improvement.
Best wishes
· INTRODUCTION: I think that the authors have well demonstrated the interest of their study. However, I think they could have explored its international scope, even if they gave as an example the case of Taiwan which they have more data about. For example, the sentence "A nursing home normally has only one escape exit, which poses a greater evacuation risk for the staff." It removes generality because, thankfully, it is not like that in many other countries and brings nothing really relevant to the aims of the article.
I think that the text from line 50 to 64 is repetitive and without an obvious interest, besides it is not complete, missing for example the lack of ability to perceive alerts.
· Maybe in Figure 4 there is a mistake. The left side, which in the legend appears as unexposed surface, is the one with the flame!
· Ln 169: "...flow rate was 111 L/min" also give the information in L/min/m2 so the methodology can be broadly used
· Ln 169-171: Why did they choose that value of water flow rate, that pressure, that velocity or that film thickness? Studies normally vary the conditions, but here they used unique conditions. Why?
· Is the film thickness at the top or at the bottom?
· What was the number of repetitions. If just one experiment, this greatly impoverish the work
· Lns 192-194/Lns 221-223/Fig 8 - I would like to have these results in the same detail for the tests without water. This is the only way to effectively compare whether the heterogeneity is solely due to water distribution.
· Fig 7 and 8- it would make sense to have a figure with the variation of temperature (relative or absolute) as a function of time, since in Fig 7 the curves vary - if they are very close, why not present them all in a single image side by side with Fig 8?
· Lns 226/233/240 have a very repetitive text
· Is figure 9 the same of Figure 8? If the data are the same, I think Fig 8 should be removed
· You should indicate in this figure where are the thermocouples to better analyse if they interfere in the water and temperature distribution
· Ln 266: that decrease in height is not so clear in the IR images!
· Ln 269: "In addition, the water supply is the most important part of the system. In this study, the water supply was set at the minimum needed to form a uniform water film." - that's why I think an analysis should be done as a function of the volume of water
· Figure 11: please indicate what "q" and "q_fail" means
· Ln305 until Fig12: This should not come in "Results" but in introduction and methodology
· Explain why you chose this household design. Was it because a fire in the past took place there?
Author Response
Point-by-point reply to reviewers’ comment
1. INTRODUCTION: I think that the authors have well demonstrated the interest of their study. However, I think they could have explored its international scope, even if they gave as an example the case of Taiwan which they have more data about. For example, the sentence "A nursing home normally has only one escape exit, which poses a greater evacuation risk for the staff." It removes generality because, thankfully, it is not like that in many other countries and brings nothing really relevant to the aims of the article.
RESPONSE: We are very grateful to the reviewer for the valuable suggestion. In the Introduction of the revised manuscript, we deleted some sentences which will bring nothing really relevant to the aims of the article.
2. I think that the text from line 50 to 64 is repetitive and without an obvious interest, besides it is not complete, missing for example the lack of ability to perceive alerts.
RESPONSE: We are very grateful to the reviewer for the valuable suggestion. In the Introduction of the revised manuscript, The text from line 50 to 64 has been deleted.
3. Maybe in Figure 4 there is a mistake. The left side, which in the legend appears as unexposed surface, is the one with the flame!
RESPONSE: We are grateful to the reviewer for critically reviewing our manuscript and providing valuable comments. The title of Figure 4 has been corrected.
4. Ln 169: "...flow rate was 111 L/min" also give the information in L/min/m2 so the methodology can be broadly used
RESPONSE: We are grateful to the reviewer for critically reviewing our manuscript and providing valuable comments. We corrected this sentence and added the information in L/min/m2 to this sentence.
5. Ln 169-171: Why did they choose that value of water flow rate, that pressure, that velocity or that film thickness? Studies normally vary the conditions, but here they used unique conditions. Why?
RESPONSE: We are grateful to the reviewer for critically reviewing our manuscript and providing valuable comments. Regarding the reason why we selected a specific condition in our fire test. We added an explanation to the revised manuscript. Please refer to Line 149~151 and Table 2.
6. Is the film thickness at the top or at the bottom?
RESPONSE: We are grateful to the reviewer for critically reviewing our manuscript and providing valuable comments. The film thickness in Line 170 of the original manuscript is averaged film thickness. It is very difficult to measure the water film thickness at any point of curtain. Thus, we measure the volume of return water in the drain and divide it by the curtain area to estimate the average water film thickness on the surface of curtain before the fire test.
7. What was the number of repetitions. If just one experiment, this greatly impoverish the work.
RESPONSE: We are grateful to the reviewer for critically reviewing our manuscript and providing valuable comments. Since the fire and smoke-barrier test is quite expensive and time-consuming, we only conducted two tests under each condition to obtain the reproducibility of the experimental results. This statement has been added in Line 233~235.
8. Lns 192-194/Lns 221-223/Fig 8 - I would like to have these results in the same detail for the tests without water. This is the only way to effectively compare whether the heterogeneity is solely due to water distribution.
RESPONSE: We are grateful to the reviewer for critically reviewing our manuscript and providing valuable comments. We strongly agree that it is necessary to add the fire test results for the curtain without a water film system. Please refer to subsection 3.1. Heat Resistance Experiment for the fire retardant curtain without a water film system in the Results and discussions in the revised manuscript.
9. Fig 7 and 8- it would make sense to have a figure with the variation of temperature (relative or absolute) as a function of time, since in Fig 7 the curves vary - if they are very close, why not present them all in a single image side by side with Fig 8?
RESPONSE: We are grateful to the reviewer for critically reviewing our manuscript and providing valuable comments. We combined Figure 7 and Figure 9 into a new figure, for the convenience of the reader to compare the temperature variation in the furnace and on the unexposed surface of the curtain over time.
10. Lns 226/233/240 have a very repetitive text
RESPONSE: We are very grateful to the reviewer for the valuable suggestion. These sentences have been rephrased. Please refer to Line 286~301 in the revised manuscript.
11. Is figure 9 the same of Figure 8? If the data are the same, I think Fig 8 should be removed
RESPONSE: We are very grateful to the reviewer for the valuable suggestion. The data in Fig. 8 is the same as that in Fig.9. Fig. 8 has been removed in the revised manuscript.
12. You should indicate in this figure where are the thermocouples to better analyse if they interfere in the water and temperature distribution
RESPONSE: We are grateful to the reviewer for critically reviewing our manuscript and providing valuable comments. We strongly agree that the thermocouples mounted in the fire test could interfere with the water film uniformity and lead to the measured temperature cannot reflect the real thermal distribution on the unexposed surface of the curtain. According to our experimental results, the critical thermocouple measurement position for evaluating the fire resistance performance of a curtain is that at the lower height of the curtain with a water film system. The water film thickness at the upper position is thicker than that at the lower position. Thus, the measured temperature at the upper position is always lower than that at the lower position.
13. Ln 266: that decrease in height is not so clear in the IR images!
RESPONSE: We are grateful to the reviewer for critically reviewing our manuscript and providing valuable comments. Actually, it is not easy to judge the water film thickness via IR thermal imager, we deleted this statement in the manuscript.
14. Ln 269: "In addition, the water supply is the most important part of the system. In this study, the water supply was set at the minimum needed to form a uniform water film." - that's why I think an analysis should be done as a function of the volume of water
RESPONSE: We are grateful to the reviewer for critically reviewing our manuscript and providing valuable comment. We added an explanation to the revised manuscript. Please refer to Line 149~151 and Table 2.
15. Figure 11: please indicate what "q" and "q_fail" means
RESPONSE: We are grateful to the reviewer for critically reviewing our manuscript and providing valuable comments. The meaning of q and q_fail are added in Line 332~334.
16. Ln305 until Fig12: This should not come in "Results" but in introduction and methodology
RESPONSE: We are grateful to the reviewer for critically reviewing our manuscript and providing valuable comments. This part from Ln305 until Fig12 in the original manuscript has been moved to Materials and Method: 2.2 Evaluation of the Application of a Fireproof Curtain with the Water Film System in a Nursing Home.
17. Explain why you chose this household design. Was it because a fire in the past took place there?
RESPONSE: We are grateful to the reviewer for critically reviewing our manuscript and providing valuable comments. The selected nursing home is a typical nursing home in Taiwan. The reason we choose this building is that it does not pass fire regulations and has a relatively large fire risk. We would like to investigate the fire safety performance enhancement of such a building by using a fire retard curtain with a water film system.
Round 2
Reviewer 3 Report
Dear Authors
Thank you for accepting all the suggestions I made regarding the previous version. I believe that the paper is now better and in a condition to be published. I would however draw attention to two aspects that are so minor and can be included in the editorial proofs and do not need to go for minor revisions:
- Table 2: "2.9" should have three significant figures and "81.2" should have two, in order to be in line with the other values of the respective columns
- In figure 12, the meaning of "q", "q_fail" and "Pm" should appear in the figure or in the legend since figures should be self explanatory.
Good work